# System Comparison for Gait and Balance Monitoring Used for the Evaluation of a Home-Based Training

**DOI:** 10.3390/s22134975

**Published:** 2022-06-30

**Authors:** Clara Rentz, Mehran Sahandi Far, Maik Boltes, Alfons Schnitzler, Katrin Amunts, Juergen Dukart, Martina Minnerop

**Affiliations:** 1Institute of Neuroscience and Medicine (INM-1), Research Centre Juelich, 52428 Juelich, Germany; k.amunts@fz-juelich.de (K.A.); m.minnerop@fz-juelich.de (M.M.); 2Institute of Neuroscience and Medicine, Brain & Behaviour (INM-7), Research Centre Juelich, 52428 Juelich, Germany; m.sahandi.far@fz-juelich.de (M.S.F.); juergen.dukart@gmail.com (J.D.); 3Institute of Systems Neuroscience, Medical Faculty, Heinrich-Heine University Duesseldorf, 40225 Duesseldorf, Germany; 4Institute for Advanced Simulation (IAS-7), Research Centre Juelich, 52428 Juelich, Germany; m.boltes@fz-juelich.de; 5Department of Neurology, Center for Movement Disorders and Neuromodulation, Medical Faculty, Heinrich-Heine University Duesseldorf, 40225 Duesseldorf, Germany; schnitza@med.uni-duesseldorf.de; 6Institute of Clinical Neuroscience and Medical Psychology, Medical Faculty, Heinrich-Heine University Duesseldorf, 40225 Duesseldorf, Germany; 7C. and O. Vogt Institute for Brain Research, Medical Faculty, University Hospital Duesseldorf, Heinrich-Heine University Duesseldorf, 40225 Duesseldorf, Germany

**Keywords:** gait, balance, training, biomarkers, motion capturing, smartphone, IMU, video-based, home-based

## Abstract

There are currently no standard methods for evaluating gait and balance performance at home. Smartphones include acceleration sensors and may represent a promising and easily accessible tool for this purpose. We performed an interventional feasibility study and compared a smartphone-based approach with two standard gait analysis systems (force plate and motion capturing systems). Healthy adults (*n* = 25, 44.1 ± 18.4 years) completed two laboratory evaluations before and after a three-week gait and balance training at home. There was an excellent agreement between all systems for stride time and cadence during normal, tandem and backward gait, whereas correlations for gait velocity were lower. Balance variables of both standard systems were moderately intercorrelated across all stance tasks, but only few correlated with the corresponding smartphone measures. Significant differences over time were found for several force plate and mocap system-obtained gait variables of normal, backward and tandem gait. Changes in balance variables over time were more heterogeneous and not significant for any system. The smartphone seems to be a suitable method to measure cadence and stride time of different gait, but not balance, tasks in healthy adults. Additional optimizations in data evaluation and processing may further improve the agreement between the analysis systems.

## 1. Introduction

Gait and balance are impaired in aging, but also in various orthopedic and in particular neurological disorders. This impairment is often associated with reduced walking speed, increased gait variability or increased postural sway [1,2,3] and can lead to considerable constraints in daily life (e.g., bradykinesia/akinesia and freezing of gait in Parkinson’s disease [4], unstable and wide-based gait in ataxias [5,6]). Identifying and assessing these constraints in daily life and providing suitable therapeutic (training) options such as physiotherapy is highly important.

There is an increasing demand to monitor physiological functions and disease-related symptoms independent of the physical presence of the respective participants or patients at the study site. Enabling study participation in a home-based setting, e.g., for human physiological monitoring [7,8] or by using wearables as measurement devices for assessing gait and balance [9,10], is an interesting and promising approach. Inertial measurement units (IMUs) consisting of accelerometers, gyrometers and magnetometers are routinely embedded in the hardware of smartphones. Due to their broad availability and the convenient option to implement applications, they may provide an attractive hands-on tool for measuring gait and balance in home-based settings. However, this set-up has been applied only recently in the field of motion analyses [10,11] and is not yet part of the standard clinical tools of measuring gait and balance.

Currently, the most commonly used instruments for gait and balance analysis are force plates (pressure-sensitive walkways) and body-worn motion capturing (mocap) systems based on IMU or optical data [12]. All these stationary systems allow the detection of abnormal or altered gait patterns in various neurological disorders such as Parkinson’s disease (PD, [13]), multiple sclerosis (MS) or ataxias [14]. Using force plates (GAITRite, 5.1 m), it was shown that PD patients have a longer stride duration, a shorter stride length and greater variability in both, compared to healthy controls [13]. In addition, stride length and velocity were reduced in ataxia patients (force plate and body-worn sensors; [14]). These gait analysis systems were also able to detect performance changes after interventions. For example, Conradsson et al. [15] found improved gait velocity and stride length in normal gait after a ten-week balance training in PD patients. They measured normal walking with a GAITRite 9 m electronic walkway with and without a cognitive task and used the averaged result of six trials. Similarly, Giardini et al. [16] used the averaged results of four trials of normal walking at usual speed on a GAITRite 4.5 m electronic walkway and showed that two forms of physical exercise training (balance exercises and mobile platform training) improved gait speed in patients with PD, whereas only the balance exercises led to improved cadence and stride length.

Although the completion time of the Timed-Up-and-Go (TUG) test is used as a standard for quantifying functional mobility in a clinical context [17], electronic assessment of balance has increasingly been used in research [18]. The most commonly used instrument is a force plate (similar to gait analysis), however, an increasing number of technologies, whose reliability and validity was described in Baker et al. [19], are being used on a regular basis (e.g., inertial sensors). For balance tasks, center of mass or center of pressure data are commonly used to determine the area of postural sway, path length and mean velocity [20,21]. Morenilla et al. [22] described altered sway areas and velocities in PD patients when examining normal stance on a tri-axis force plate (Kistler). They found significant increases in total sway area and in mean anteroposterior and mediolateral displacement for PD patients. Moreover, Sun et al. [23] reported that both a new inertial body-worn sensor and a force plate were able to discriminate between subjects with severe MS and healthy control. However, only the force plate was able to distinguish subjects with mild MS from healthy control and patients with severe MS. Studies using force plates were also able to detect changes in performance after training interventions [24,25], i.e., patients with chronic stroke showed improved sway distance after participation in a virtual reality reflection therapy [26], and children with cerebral palsy showed decreased sway area and sway path after 12 weeks of training with a gaming balance board [27].

Thus, these stationary analysis systems of gait and balance are obviously able to detect performance differences between different groups in addition to shifts in performance over time or after intervention. They stand out in terms of their accuracy and ease of use. However, whether this also holds true for smartphone-based evaluation of gait and balance is still a topic of intensive research. In contrast to force plates and whole-body IMUs, the smartphone relies on a single sensor estimating velocity from acceleration and, in addition, gravitational influences and high-frequency noise must be filtered out. The advantage of smartphones would lie in their high disposability and saving of resources. Here we compared smartphone-based assessment of gait and balance tasks before and after three weeks of training to two commonly applied stationary gait analysis systems. We evaluated the feasibility of this approach to draw conclusions about the agreement of the three gait analysis systems and their ability to detect changes after a training intervention.

## 2. Materials and Methods

In this interventional feasibility study, smartphone-based evaluation of gait and balance was combined with two common stationary gait analysis systems requiring a laboratory environment: a zebris force plate and a Xsens mocap system with inertial sensors. Overall, 25 participants were recruited into the study. Two applications (apps, “*JTrack EMA*” and “*JTrack Social*”) were installed on the smartphones of the participants (screenshots are available in Far et al. [28]). Both apps were developed at the Forschungszentrum Jülich [28]. *JTrack EMA* was developed for collection of ecological momentary assessments, so that common clinical questionnaires can be easily implemented into the app. *JTrack Social* was developed for customizable gathering of sensor data, including accelerometer information, using sensors embedded in any modern smartphone. Data were collected during a three-week video-based training intervention, which was performed at home and included twelve gait and balance training sessions, each lasting 20 min (see Figure 1). Participants were asked to indicate how many of the training videos they performed in total. Nevertheless, no verification of this information could take place. The present study was a feasibility study of a combined assessment and training protocol for gait and balance in healthy subjects. Written informed consent was obtained by all participants. The study was approved by the ethics committee of the Psychology faculty of the Heinrich Heine University Düsseldorf.

### 2.1. Participants

Twenty-five participants were recruited via notices at universities, supermarkets and social media, and via newspaper. Participants had to be aged between 20 and 70 years, needed to walk safely without a walking aid, and did not report joint problems (osteoarthritis, endoprostheses) or other neurological, muscular or other medical problems affecting gait (e.g., falls, deep brain stimulation).

### 2.2. Gait Analysis Systems

The following three gait analysis systems (see also Figure 2) were used for assessment of gait and balance tasks in this study:The zebris FDM force plate (4.24 m, zebris Medical GmbH, Isny, Germany, https://www.zebris.de/en/medical/stand-analysis-roll-analysis-and-gait-analysis-for-the-practice, accessed on 30 June 2022) with the Noraxon^®^ myoPressure software (Noraxon U.S.A., Inc., Scottsdale, AZ, USA, https://www.noraxon.com/our-products/myopressure/, accessed on 30 June 2022). This uses capacitive pressure sensors to capture the pressure distribution in gait and balance.The Xsens mocap system consists of the MVN Awinda hardware and MVN Analyze software (Xsens Technologies B.V., Enschede, The Netherlands, https://www.xsens.com/motion-capture, accessed on 30 June 2022). It consists of 17 IMUs attached to each distinctive segment of the body fixed with body straps, which record angular velocity, acceleration, atmospheric pressure and the Earth’s magnetic field with a frequency of 60 Hz.Individual Android-based smartphones of the participants on which the JTrack Social app was installed [28]. During all measurements, the accelerometer data of the smartphone were recorded using this app. The smartphone was placed in a waist bag.

#### 2.2.1. Force Plate Feature Extraction

The zebris FDM *force plate* uses capacitive pressure sensors to capture the pressure distribution in gait and balance. No preprocessing was performed on the force and pressure data, which were recorded with a frequency of 100 Hz. Gait or balance reports are created automatically in the Noraxon myoPressure™ software, by selecting “Report” → “Bilateral Gait Report” for gait tasks and “Report” → “Stance Report” for stance tasks. The software uses the vertical ground reaction force to determine gait phases such as the heel strike or toe off. Movements in the beginning and at the end of the tasks that were not part of the task were unselected for all tasks. Apart from this, the entire distance walked on the force plate was included in the analysis. Feet positions were checked manually for tandem gait, since the software frequently was not able to distinguish the order of the left and right feet in this task. If foot positions were wrong according to the synchronized video, they were switched manually (left feet contacts were exchanged for right feet contacts). 

In the report, stride time (s) describes the time between two heel contacts on the same side of the body. Cadence is the number of steps performed per second. The average velocity calculated for the force plate is the average stride length divided by the average stride time. Step width (cm) is the lateral distance between the center of the left and right heel.

#### 2.2.2. Mocap System Feature Extraction

The Xsens *mocap system* computes the full-body motion based on constraints from a biomechanical model of the human skeleton with the help of sensor fusion algorithms. To configure the biomechanical model, body dimensions such as foot length, hip height and shoulder width of each participant were collated. The attached IMUs of the system are self-contained and light weight, so that they do not restrict subjects in their freedom of movement. After placing the system on a participant, a calibration process was performed as described in the MVN User Manual [29], i.e., to calculate the orientations of the sensors with respect to the corresponding segments. Quantities regarding the accuracy of the tracker and the MVN fusion engine can be found in the MVN User Manual [29]. A detailed description of the system is given in Schepers et al. [30].

The data were recorded with the Xsens MVN 2020.2 software and stored in the mvnx format after reprocessing in HD. A Python script was used to extract the position of the pelvis and both feet (foot segments located between the ankles within the Xsens model, see section 23.6.10 in the MVN User Manual [29]). The pelvis data were used to approximate the center of mass (COM, sensor position at the lower back on top of the sacrum). Data are given in the x-direction (anterior–posterior), in the y-direction (medial–lateral) and in the z-direction (vertical). The definition of axes also applies to the data of the left and right foot. The following procedures were separately repeated for each participant and each task.

Data were visualized to check for plausibility and to avoid including errors. Since the data contained turns at the end and at the beginning (most anterior and most posterior points, *x*-axis) of each lane, the first and last meters in the x-direction were excluded from the data. Data were then split into separate lanes (6 lanes for normal gait, 6 lanes for backward gait, 4 lanes for tandem gait) that every participant walked. IMU sensors showed a drift after a few lanes of walking, resulting in a mismatch between the correct direction of travel and the sensor-based detected direction of the *x*-axis as the main walking direction. This was corrected by rotating the data within the moving plane (x–y) to maximize the conformance between the walking direction and the *x*-axis. To calculate the time between two consecutive steps of the participant (step time), the vertical component of the COM data was used. As the COM moved up and down in cyclic movements, its peaks were used as markers for a step cycle. The height to find the peaks (scipy.signal, find_peaks) was adapted for each participant by visually checking the output plots. To avoid technical errors and enable single step detection during the tandem gait, an individual minimum distance between two consecutive peaks was required. The time between two steps (inter-step time) was calculated by subtracting the times of two neighboring peaks. 

The step frequency (cadence), defined as the number of steps per second, is the inverse of the inter-step time.

Velocity as distance per time was calculated separately for each lane using the difference between the first and the last data point for position and time. 

To calculate the lateral distance between both feet during steps (step width), the vertical *z*-axis and the *y*-axis (medial–lateral displacement) of the feet were considered. The time frame with the lowest foot position of each foot (mid-stance phase) was marked by searching for the minima in the z-direction (vertical axis). Its position in the y-direction at the same time frame was used to determine the distance between the left and right feet. Height and width in the find_peaks function were again adapted individually for each participant. 

For the balance tasks, data import and inspection were performed in a similar way as described for the gait tasks. For each participant, the time span for analysis was selected in a way such that movements in the beginning or at the end of the balance task were excluded. Analysis was performed on the pelvis data (COM). The total path length that was traveled by the COM of the participant was calculated by summing the distance between all successive points in the path within the moving plane (x–y). The sway velocity described the number of millimeters the COM of the participant moved per second.

The area of an ellipse around the COM path was calculated by multiplying the anteroposterior sway and the mediolateral sway with pi. 

#### 2.2.3. Smartphone Feature Extraction

The JTrack Social app was installed on the individual Android-based smartphones of the participants and placed in a waist bag during the measurement (placed at the lower belly to approximate the COM while also ensuring simple handling). 

All analyses of the JTrack Social app data were performed in MATLAB. The accelerometer data for each smartphone were recorded using the highest frequency provided for the respective smartphone (the recorded frequencies ranged between 100 and 252 Hz). All recorded gait and balance data were visually quality checked by removing non-tasks and, where identifiable, turn periods from the recordings. For normal gait data, manual step labeling was performed to obtain reference data for automated step labeling using a dedicated open-source MATLAB toolbox implemented for that purpose (https://github.com/juryxy/step_detector, accessed on 30 June 2022). 

All accelerometer data were band-pass filtered in the range of 0.8–20 Hz to remove the gravitational component and the high frequency noise. Step detection for gait data was performed using the findpeaks function on the Euclidean norm of the accelerometer data. For this function, the following two parameters can be optimized for step detection—the minimum peak height (further expressed as standard deviation (SD) relative to the mean signal) and the minimum peak distance (in seconds). As the zebris FDM force plate was able to directly capture steps using pressure sensors, it was considered as closest to the ground truth together with the manually labeled data for normal gait. To identify optimum parameter combinations for smartphone step detection, we performed a grid search for the above parameters (peak height: 1.5 SD in steps of 0.1 to 3.0 SD; peak distance: 0.2 s in steps of 0.02 to 0.44 s), testing for correlations between the mean stride intervals (MSIs) obtained using these settings and MSIs derived using the ground truth provided by the force plate and manual labeling (Figure A1, Appendix A). For normal and backward gait, the optimum parameters providing the closest overall correlation to the ground truth were a minimum peak height of 2.3 SD and minimum peak distance of 0.38 s. For tandem gait, the optimum peak height was 2.7 SD and minimum peak distance was 0.42 s. Using these optimum parameters for step detection, the following features were computed using dedicated MATLAB scripts: stride time, cadence and velocity. To compute the mean velocity, we performed a step-wise double integration of accelerometer data to velocity and displacement using the first point as a reference. Thereby, the above band-pass filter was re-applied at each step to ensure that the residual gravitational and potential reintroduced high-frequency effects were removed from the data. Mean velocity (in m/s) was then computed as distance covered during the gait tasks divided by time. 

For stance tasks, accelerometer data were transformed into displacement. The gravitational and high-frequency components were removed from acceleration and displacement data using band-pass filtering as for the gait tasks. Mean velocity was computed as point-by-point displacement divided by time. As the smartphone had no specific fixation of the phone orientation (except for a waist bag), the orientation of sensors with respect to the x- and y-plane differed across phones. To obtain an estimate of postural sway, we therefore performed a principal component analysis to determine the main directions of the sway in the three-dimensional space. The ellipsoid volume encompassing the 95% confidence interval of all points across the three principal components was computed as an estimate of postural sway around the COM (Figure A2, Appendix A).

An additional app, the JTrack EMA app (Biomarker Development, INM-7, Forschungszentrum Jülich), was used for the retrieval of questionnaires.

### 2.3. Study Tasks

#### 2.3.1. Gait and Balance Tasks

For all gait tasks, participants were asked to walk safely across the force plate, then turn around behind the plate and walk back to the starting position. The walks were repeated several times with the number of iterations varying between tasks (for details see Table 1). For tandem gait, participants walked in a straight (imaginary) line by placing one foot in front of the other, placing the heel of one foot about a hand’s width in front of the toes of the previous foot to enable separate foot detection by the force plate software. In the balance tasks, the participants were asked to keep their balance for as long as possible without leaving their position or holding up (maximum of 30 s). Participants performed all tasks without wearing shoes.

#### 2.3.2. Questionnaires

Age, gender, body height, body weight, profession and years of education were retrieved in a demographic questionnaire during the first laboratory visit. To assess depression and anxiety, the German versions of the depression module of the patient health questionnaire (PHQ-9 [31], German version: [32]) and the hospital anxiety and depression scale ([33], German version: HADS-D [34]) were used. Additionally, general habitual well-being (FAHW [35]) and self-efficacy, optimism and pessimism (SWOP-K9 [36]) were assessed. To assess self-efficacy in relation to falls, the (modified) German version of the Activities-Specific Balance Confidence scale was used (ABC-D [37]).

The “PHQ_stress” and “PHQ_depression” subscores were selected from the PHQ-9 questionnaire. Although the depression and anxiety variables were used as exclusion criteria, the stress variable ranged from 0 to 20 and served as a covariate to describe the population. The anxiety subscore of the HADS-D had a cut-off value of >10 points and a depression subscore of >8 points. In the FAHW score, a total score of 38 to 50 or 35 to 47 (men and women, respectively) was defined as “average” according to the authors of the questionnaire. Additionally, the score contains a row of “smiley” icons, ranging from a happy face to a sad face. This was included in the evaluation by assigning a 1 to the happiest smiley and a 7 to the saddest smiley. The SWOP-K9 questionnaire contained items on self-efficacy (SWOP-SE), optimism (SWOP-OP) and pessimism (SWOP-PS), with scores ranging from 5 to 20, 2 to 8 and 2 to 8, respectively. For the ABC-D questionnaire, the scale was adapted to a 4-point response scale (not confident at all, somewhat less confident, somewhat confident, absolutely confident) so that a score between 16 (maximum confidence) and 64 (minimum confidence) could be achieved.

#### 2.3.3. Training at Home

Gait and balance training was performed four times per week for 20 min by instruction via provided videos. The videos were produced by a physical therapy practice (PhysioStützpunkt, Köln, Germany) and uploaded to Vimeo (https://vimeo.com/, accessed on 30 June 2022). In each video, an experienced physiotherapist explained and demonstrated various tasks to improve gait and balance and instructed the participants to follow along. This included strength training, coordination training, stability training and mobility. The twelve videos progressed from simple to more demanding tasks and also included suggestions to reduce or increase the level of difficulty. Videos could be paused or repeated at any time, but participants were instructed to perform each training session only once until their second study visit was completed.

### 2.4. Statistical Analyses

From the set of extractable variables of each gait analysis system and each gait task, three variables were selected that were consistently available across all systems (see Table 2): *Gait velocity* (average velocity across all straight distances covered in the task, measured in meters per second), *stride time* (average duration of one stride defined as two consecutive steps in seconds) and *cadence* (average number of steps that are performed within one second). Additionally, *step width* was extracted from the force plate gait report and from the mocap system data, as this is an important variable to detect abnormal gait patterns (e.g., broadened base of support in cerebellar ataxias, see [3]). However, the step width cannot be derived from the acceleration data of the smartphone and was therefore not extracted from the smartphone data. For the balance tasks, the center of mass *(COM) sway area* (area of an ellipse enclosing all data points in the x- and y-direction) and the *velocity of the COM* (average distance in millimeters that the participant traveled per second) were chosen. These two variables showed good reliability in previous studies (e.g., [38,39]) and are commonly used for examining balance performance [20,21,40]. Both variables were available for all three gait analysis systems.

Correlations between the questionnaire scores, between the individual variables *within one* gait analysis system, and between variables in *all* gait analysis systems, were calculated with the Pearson correlation coefficient. In this context, a correlation between 0.10 and 0.39 was described as weak, 0.40 to 0.69 as moderate and 0.70 to 1.00 as strong [40]. To analyze changes over time between the questionnaire scores and gait and balance variables at the first and second study visit (T1 and T2), either an ordinary paired-sample *t*-test was performed if the data scores were normally distributed, or a Wilcoxon rank test, if the data were not normally distributed. For all statistical analyses, a *p*-value of <0.05 was considered significant. Since results were corrected for multiple comparisons using a Bonferroni correction, the resulting *p*-values of <0.013 (force plate, mocap system) and <0.017 (smartphone) were considered significant when reporting changes over time. Boxplots of all gait and balance variables were checked and extreme outliers were excluded (>3 ∗ IQR above quartile 3).

## 3. Results

### 3.1. Participants

A total of 25 participants (age 44.0 ± 18.4 years) took part in the first study visit (T1, 52% female, 92% right-handed, see Table 3). One participant had missing data from the mocap system due to technical problems.

For the second study visit, four participants dropped out (injury independent of the study (one), technical difficulties (one) and time constraints (two)). This led to a sample of 21 participants at T2 with an average age of 44.7 ± 19.4 years (57% female, 95% right-handed). All subjects reported having performed each of the training videos (12/12).

All demographic variables and questionnaire scores except the ABC-D score were normally distributed. Because one participant showed a depressive mood (HADS-depression score 10), all analyses were conducted with and without this subject. Since results did not differ, data from this participant were not excluded from further analyses.

Of the gait and balance variables, 8 of 33 gait variables were not normally distributed and 21 of 24 balance variables were not normally distributed. Accordingly, non-parametric statistical tests were selected for these variables. For detailed specifications of the variables, please see Table A2 (Appendix A).

### 3.2. Questionnaires

No differences between the questionnaires obtained at both study visits were found between T1 and T2 (Table 4, *p* > 0.09).

### 3.3. Gait and Balance Performance

#### 3.3.1. Conformity of the Systems

Significant correlations between corresponding gait variables (stride time, cadence, velocity) across the three systems were present during all gait tasks. For the velocity variable during the backward and tandem gait, the correlations involving the smartphone were weak and did not all reach significance; correlations for the other two variables were significant.

For normal gait (Table 5), strong correlations were found between the three corresponding gait variables (stride time, cadence, velocity) of the force plate, mocap system and smartphone, except for one moderate correlation of velocity between the mocap system and smartphone. Step width was moderately correlated between the force plate and mocap system.

For backward gait (Table 6), strong correlations were found between the stride time variables of all systems and for cadence between the force plate and smartphone. The remaining correlations regarding cadence and velocity were moderate or even showed no correlation for velocity between the mocap system and smartphone.

For tandem gait (Table 7), correlations were again strong between stride time and cadence variables across all three systems. However, for velocity, only moderate correlation was found between the force plate and the mocap system, but not between the smartphone and the two standard systems. 

For balance tasks, moderate to strong significant correlations were found between the corresponding variables of the force plate and mocap system (see Table 8). For smartphone data, only three variables reached statistical significance (moderate correlations between the ellipse variables in tandem stance and the velocity variables in narrow stance with eyes closed between the force plate and smartphone, and a moderate correlation between the velocity variables in single leg stance between the mocap system and smartphone).

#### 3.3.2. Reference Values

To put the outcome values of the gait tasks in context, reference values from the literature are given in Table 9.

#### 3.3.3. Differences over Time—Force Plate

Since not all variables were normally distributed, *p*-values either refer to *t*-tests (no indication) or to Wilcoxon-rank tests (indicated by “(W)”). 

For normal gait, a significant difference was found in all variables between T1 and T2: stride time (*p* = 0.003, Figure 3A), cadence (*p* = 0.002, Figure 3B), velocity (*p* = 0.002, Figure 4A) and step width (*p*(W) = 0.004, Figure 4B). For the backward gait, only the velocity variable (*p* = 0.005, Figure 4A) remained significant after correcting for multiple comparisons. For tandem gait, none of the variables remained significant after correcting for multiple comparisons.

For the stance tasks, none of the variables remained significant after correcting for multiple comparisons (Figure 5A and Figure 4B).

The exact values for all tasks and gait analysis systems are reported in Table A1, Appendix A.

#### 3.3.4. Differences over Time—Mocap System

In contrast to the force plate, a significant difference in normal gait was found in only two of four variables: stride time (*p* = 0.002, Figure 3C) and cadence (*p* = 0.001, Figure 3D). For the backward gait, only the velocity variable (*p* = 0.007, Figure 4C) remained significant after correcting for multiple comparisons—similar to the results of the force plate. For the tandem gait, a significant difference was found for two of four variables: for the stride time (*p* = 0.003, Figure 3C) and the cadence (*p* = 0.001, Figure 3D). No significant effect was found for the step width (Figure 4D); however, this may be related to the initial calibration procedure: the closer the participants’ feet were in the “neutral position”, the smaller the absolute values of the step width were in the later analysis. 

Similar to the force plate, the mocap system analysis did not reveal a significant difference between T1 and T2 for any of the stance tasks.

#### 3.3.5. Differences over Time—JTrack Smartphone Platform

In contrast to both the force plate and mocap systems, none of the variables of normal gait, backward gait or tandem gait remained significant after correcting for multiple comparisons (Figure 3E,F and Figure 4E). Compared to the other gait analysis systems, the smartphone had a much higher variability of the velocity values, e.g., velocity values of the backward gait at T1 were 0.69 ± 0.09 m/s for the force plate and 0.55 ± 0.43 m/s for the smartphone (see Table A1, Appendix A).

Similar to both the force plate and mocap systems, the smartphone analysis showed no significant differences between T1 and T2 for any of the stance tasks (Figure 5E,F).

## 4. Discussion

Here, we performed an interventional feasibility study and compared three systems for the monitoring of home-based gait and balance training in healthy adults. In particular, we assessed the applicability of smartphone-based data collection in comparison to standard methods and the capability of the methods to detect performance changes after training. 

### 4.1. Conformance of the Three Gait Analysis Systems

Gait variables obtained with both standard analysis systems (force plate and mocap) showed moderate to strong intercorrelations, except for step width. However, the strength varied depending on the performed gait task with excellent correlations for normal gait. Step detection during backward or tandem gait was more challenging and error-prone compared to normal gait, since feet were placed more cautiously and slowly, resulting in lower force and acceleration values, in addition to atypical movement patterns. In line with this, step width values correlated moderately between both systems for normal but not for backward gait. For tandem gait, the correlation between the step width values of both systems even revealed negative values, due to the calibration process of the mocap system [29]: if participants placed their feet in a very narrow stance during the “neutral position”, required for the calibration process, the absolute values of the step width were much lower in the later analysis. This led to incorrect lateral positions of the feet and even to negative step width values in the tandem gait. For future studies using mocap systems, a standardized stance position of the participants is therefore highly recommended. 

The JTrack based smartphone evaluation using accelerometer data showed strong correlations for the stride time and cadence variables of all gait tasks with both standard systems. Velocity, however, showed only moderate to strong correlations for normal and backward gait, and weak correlations for tandem gait. Taken together, all three gait analysis systems showed excellent agreement during normal gait, followed by the tandem gait task and a substantially lower agreement for the backward gait task. The agreement was better for the gait variables of stride time and cadence than for velocity. The less accurate velocity estimation via smartphone relied on a single sensor estimating velocity from acceleration using the first recorded value as a reference. As this first value was not calibrated in our study (i.e., no fixed position was taken of the phone when recording started), this may lead to biases in estimation of the initial velocity. It also explains the lack of correlation with other systems for tandem gait, for which the velocity was substantially lower, thereby increasing the impact of noise.

The strong correlations of smartphone-based gait variables with standard gait analysis systems found in our study are in contrast to Steins et al. [48], who described only moderate agreement between an iPod touch and an Xsens sensor when investigating the reliability of inertial sensors of smart devices during normal gait in healthy adults. Nevertheless, other studies suggested that smart devices are an acceptable method for assessing gait in rheumatic patients [49] and have the potential for future use in the clinic [13].

The stance variables of ellipse area and velocity showed moderate to strong correlations between the two standard force plate and mocap systems (see Section 3.3), in spite of large differences in the absolute values obtained with these methods (see Table A2, Appendix A). In contrast, only weak to moderate correlations were found between the smartphone and both other systems. This might be due to specific aspects of data acquisition and analysis. Force plates can directly register the foot print and determine the respective variables from position data. In contrast, the smartphone uses accelerometer information with respect to the first recorded value and thus only infers position data through double integration. Thereby, gravitational influences and high-frequency noise must be filtered out using band-pass filtering, which may lead to additional biases in position estimation. The mocap system uses multiple sensors, e.g., directly on the feet, and, in addition to the accelerometer data, also considers angular velocity, atmospheric pressure and magnetic field data, and a biomechanical model. This contrasts with the smartphone analyses, which relied on a single sensor near the COM. This enables the mocap system to determine the positions of the sensors relative to one another and to better estimate the gravitational and the noise components. Since the position and orientation of the smartphone were not fixed when recording started, the initial estimates may be biased, affecting all derived measures. Moreover, as the three axes in space were not fixed, it is difficult to determine an area in mm^2^ in a standardized manner. Accordingly, the ellipse volume was computed in mm^3^, introducing an additional source of variation. 

Taken together, stride time and cadence seem to be variables that are robust to measurement with a smartphone, whereas other gait and stance variables are subject to some limitations.

### 4.2. Questionnaires

Since physical activity has a significant impact on mental well-being and vice versa, the objective motor assessment in this study was accompanied by a set of questionnaires addressing different aspects of subjective participant-reported outcome measures (e.g., depression- and anxiety-related symptoms, general well-being, stress, self-efficacy, optimism, pessimism and balance confidence).

Contrary to our expectations, the questionnaire scores did not differ between the pre- and post-training study visits. Physical therapy or exercises can reduce fatigue and improve one’s emotional life [50] and mental health, in a manner that is even similar to psychotherapy. By comparison, our participants already had above-average FAHW scores at their first visit (reference values are given in [35]), indicating that the general well-being was already at a high level before the training and hence left less room for improvement. 

Due to several constraints (study duration, compliance), a three-week period was chosen as the training interval in this study. Although Mikkelsen et al. [51] reported that exercising for 15 min three times per week already reduced depressive symptoms, most studies chose a longer time period for the training program or a longer duration for each unit to maximize the effectiveness of balance training and to prevent falls [52,53]. In the more specific context of home-based training, the highest effectiveness of video-based rehabilitation programs was found after at least four weeks [54]. Nevertheless, although a higher training volume or frequency can lead to better training results, it may also reduce compliance, as the subjective cost may exceed the perceived benefit of the training. In Haines et al. [55], a drop in compliance was found after three weeks. In our study, all subjects reported having performed each of the training videos, but verification of this information was not possible, impeding a valid statement regarding compliance.

### 4.3. Gait Performance

Mean values of stride time, cadence, velocity and step width obtained in our study were comparable to those found in the literature for normal gait in healthy adults (see Table 9). Similarly, stride time and cadence values during backward gait were comparable between the literature [45] and between all three gait analysis systems. However, in our study, velocity values were 20–60% lower during backward gait compared to the literature ([45,46] measured on force plates). For step width, force plate values during backward gait were in line with the literature [45], whereas the mocap system values were lower (~29%), which is likely related to the calibration, as mentioned in Section 4.1. For tandem gait, Kronenbuerger et al. [47] reported lower cadence values in tandem gait compared to our study (~34%, see Table 9), but they used a different study setting with predetermined gait speed. Rao et al. [43] used a force plate in healthy older adults (mean age 84 years) and also found slightly lower values for cadence, velocity and step width in the tandem gait compared to our values, likely related to the age difference between both cohorts. Importantly, in a tandem gait, the heel of one foot is normally placed directly in front of the toes of the other foot. In our study, a hand’s width of space had to be left between the feet to allow the force plate to distinguish between both feet. This difference may explain the higher cadence and velocity values found in our study.

There were significant improvements for some of the variables between the pre- and post-training study visits. For normal gait, the force plate analysis revealed improvement in all gait variables after training, whereas the mocap system only revealed an improvement in two variables after training (stride time, cadence) and the smartphone did not show a significant improvement. For backward gait, an improvement was shown for the velocity variable of both force plate and mocap systems. For tandem gait, an improvement after training was found for the two variables of stride time and cadence in the mocap system only.

In the best case, all systems would have shown significant changes over time in the same variables. However, the differences between the systems may result from (a) reduced statistical power due to a lower number of valid values included in the statistical analysis (as for the smartphone data), and (b) higher variability observed for smartphone data; both of which affect the outcome of the statistical tests. Regarding the two standard systems, the force plate detected more changes in normal gait over time in healthy adult subjects undergoing a training period of three weeks. By comparison, only the mocap system detected changes in tandem gait. One reason for these differences could be that the hardware and software used for the force plate are more accurate for normal walking (because it uses position data, see Section 4.1), but had difficulties distinguishing right and left feet in the tandem gait, whereas the manual detection of steps in the tandem gait was more controllable in the mocap system analysis. Nevertheless, a general improvement in gait variables was observed across all gait analysis systems. 

The observed improvements were expected and desirable changes in terms of improved gait performance after a training intervention, and have also been described in several patient studies with various disorders such as PD [15,56] and stroke [57], or for healthy (mostly older) adults after different kinds of training [58,59,60,61,62]. 

Of note, the observed improvement between pre- and post-training visits is most probably caused by the training performed between these visits. However, a control group undergoing the measurements at T1 and T2 without any training in the interim was missing and, therefore, a learning effect cannot be entirely excluded. To confirm and substantiate the positive effects of this study, further investigation, including a control group, would be reasonable in future.

### 4.4. Balance Performance

For normal stance, mean values of balance performance (ellipse area) measured with a force plate were comparable with corresponding values of healthy adults in the literature [20,63]. Although, for narrow stance, the velocity values of our study were also comparable or slightly higher than the values of the studies cited above, the values for the ellipse area differed. This is most likely due to methodological differences regarding the calculation of this variable, which is not specified in the studies mentioned above. Pomarino et al. [63] mentioned, however, that their balance measures were averaged over the recording time. In our study, averaged ellipse area values for normal stance were 24 mm^2^, 50.7 mm^2^ and 3.3 mm^3^ (force plate, mocap system and smartphone, respectively), which again is comparable to or slightly lower than in the studies by Nusseck and Spahn [20] and Pomarino et al. [63], who measured with force plates. 

For the other stance tasks, reference values for healthy adults in the literature are scarce. One study reported an ellipse area of 138 mm^2^ for the single leg stance in a control group of older adults [64], whereas we found values of 878 mm^2^, 3860 mm^2^ and 384 mm^3^ in our study (averaged values per second: 29 mm^2^, 129 mm^2^ and 13 mm^3^). However, it is unclear if the values were indeed averaged in the cited study. If so, the values in our study were lower compared to those in the literature, possibly due to a lower mean age of the participants. Values for the velocity balance variable were only reported separately for mediolateral and anteroposterior directions [64] and are thus not comparable to our values. Terra et al. [38] examined the same stance tasks we used in PD patients, using a force plate, and described an increase in the values for the COM ellipse area and velocity with the level of difficulty of the respective stance tasks, ranging from narrow stance to narrow stance with eyes closed, followed by tandem stance and, finally, single leg stance. This is consistent with our results regarding the velocity variable obtained with the force plate, whereas, for the other systems, the order of the stance tasks varied (see Figure 5).

Regarding the training effects, the statistical analysis did not show a significant improvement in balance performance between pre- and post-training measurements from T1 to T2 (see Figure 5). In contrast to the gait tasks, where small improvements in performance were observed for all variables (even though not always reaching statistical significance), the pattern of observed changes in stance tasks was more heterogeneous (see Table A1, Appendix A). In contrast, an improvement was reported in the literature for different patient groups, e.g., for PD patients [65] or for children with cerebral palsy [21,27] and healthy older adults [66], and for younger adults [67] after a training intervention. Cadore et al. [68] also summarized in their review that most balance trainings in older adults with physical frailty led to enhancements in balance. However, methods, outcome measures and training interventions were highly heterogeneous among the cited studies, impeding their comparability. 

### 4.5. Summary

Agreement between the three gait analysis systems was higher for gait variables than for balance variables. With the exception of the step width variable, both standard methods showed an excellent agreement between the values of the analyzed gait variables, especially for the normal gait task, followed by tandem and backward gait tasks. In particular, for the stride time and cadence variables, values obtained with the smartphone showed a strong correlation with values obtained with both standard systems, whereas correlations for the gait velocity variable were considerably weaker, especially for tandem and backward gait. Improvements (by percentage change) were consistently visible across all gait tasks and all three applied gait analysis systems. However, significant changes over time were only found for gait variables obtained from the force plate and mocap systems. In contrast, changes in balance variables over time yielded a highly heterogeneous pattern without clear improvement across stance tasks and applied systems. Furthermore, participant-reported outcome measures did not reveal any changes over time, which may be due to the already high level of “general well-being” at the study onset.

According to the results of our research, there is a high level of agreement between the devices used in the laboratory and smartphones. This finding is consistent with the findings of earlier studies [69,70]. The fact that smartphones and smartwatches can be put to use in everyday settings is the primary advantage of using such devices. Because of this capability, patients can be monitored in (near) real time and over extended time periods such as months and years. In addition, the vast number of people who own smartphones makes it possible to use these devices as an excellent source for crowdsourcing, regardless of the physical location of the users. However, there are additional considerations such as misunderstanding and following of instructions, effect of motivation, learning effects and misplacement or orientation of devices for at-home usage settings and self-administered protocols, both of which have the potential to affect the validity and reliability of the data collected [71].

Since improvements were found only for gait performance, the applicability of smartphones as a measurement system seems to be particularly useful in disorders in which the gait is impaired, such as PD and ataxia [13,14]. Stride time and cadence measured with the smartphone were found to have a high agreement with the measurements of the standard analysis systems and are variables that differentiate patients from healthy controls [13] or that might improve after an intervention [15]. For this reason, they seem to be eligible variables for future smartphone studies in home-based environments. Future studies should investigate the most effective intervention program and should combine a longer time frame for exercise interventions with major efforts to maintain or even improve study compliance. 

## 5. Conclusions

Our analysis showed that measuring gait and balance performance in healthy adults with wearable devices, such as smartphones, produced comparable results for the stride time and cadence variables compared to measurements with standard gait analysis systems such as the force plate or mocap systems, whereas results for gait velocity were less convincing. Potentially, adjustments may have to be made in the data evaluation for the calculation of velocity to achieve better agreement. 

Although the positive influence of three weeks of gait and balance training on gait performance in healthy adults was noteworthy, comparable improvements were found for all three gait analysis systems in gait parameters. However, only the force plate and the mocap systems were able to detect significant changes over time during the gait tasks. In contrast to the motor performance, no improvement was found for the questionnaire scores. To ensure that the improvement is indeed the effect of the training and not a test–retest effect, a further study including a control group which does not take part in a training intervention is required.

Reference values for gait and balance variables in healthy adults are currently scarce in the literature. For future analyses, the number of comparable gait and balance variables can be increased to obtain a more detailed overview of reference values of healthy adults and to compare these values with patient data (e.g., patients with movement disorders). Ellis et al. [13] also suggested that many more consecutive steps (e.g., more than 100 steps) are required to reliably detect differences in gait performance. This is not possible when using force plates with a limited length, but seems to be an interesting set-up for further smartphone-based analyses.

## Figures and Tables

**Figure 1 sensors-22-04975-f001:**
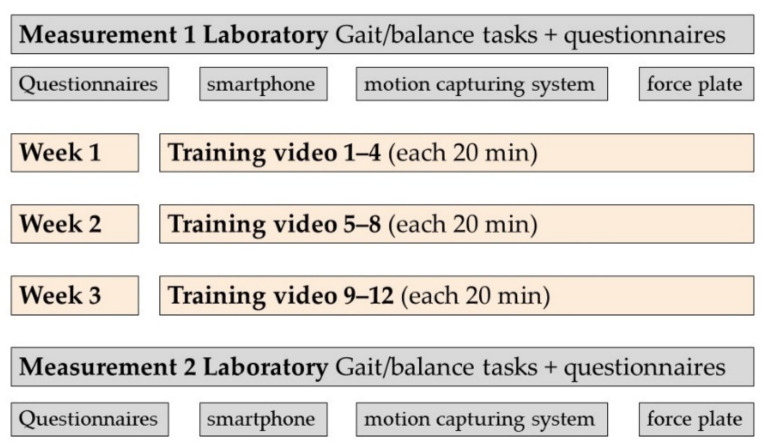
Overview of study design.

**Figure 2 sensors-22-04975-f002:**
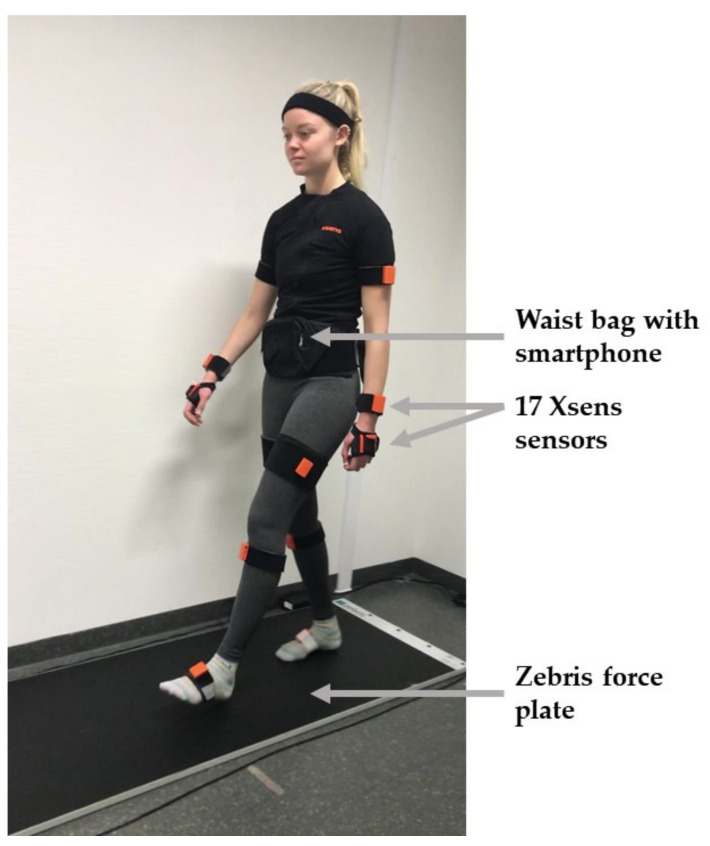
Representation of the three gait analysis systems used in the study.

**Figure 3 sensors-22-04975-f003:**
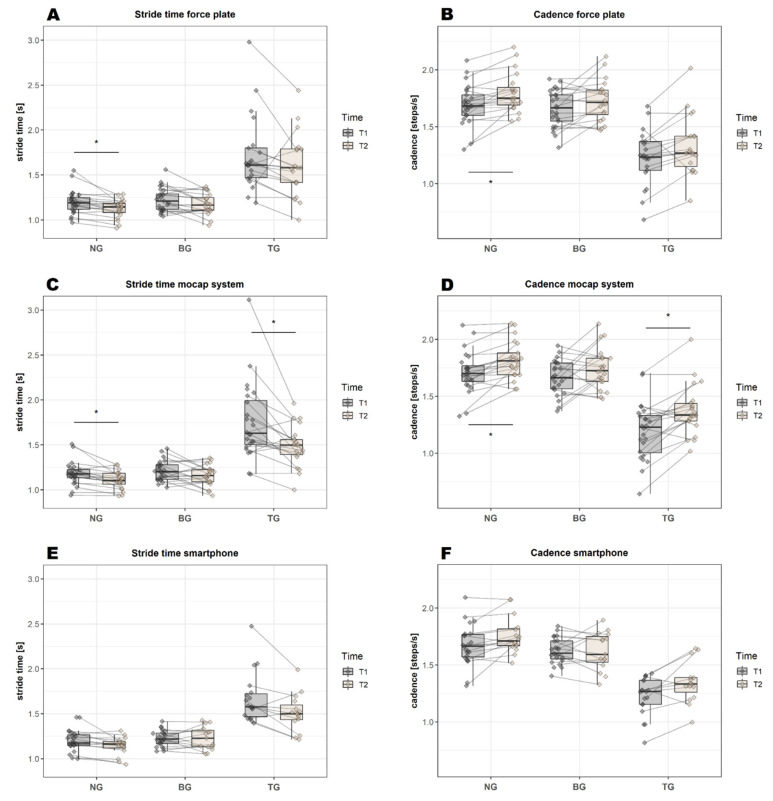
Graphical representation of the mean values of stride time and cadence for all three gait analysis systems at T1 and T2 (before and after training). Significant differences over time (after Bonferroni correction) are highlighted by an asterisk. BG = backward gait, NG = normal gait, TG = tandem gait.

**Figure 4 sensors-22-04975-f004:**
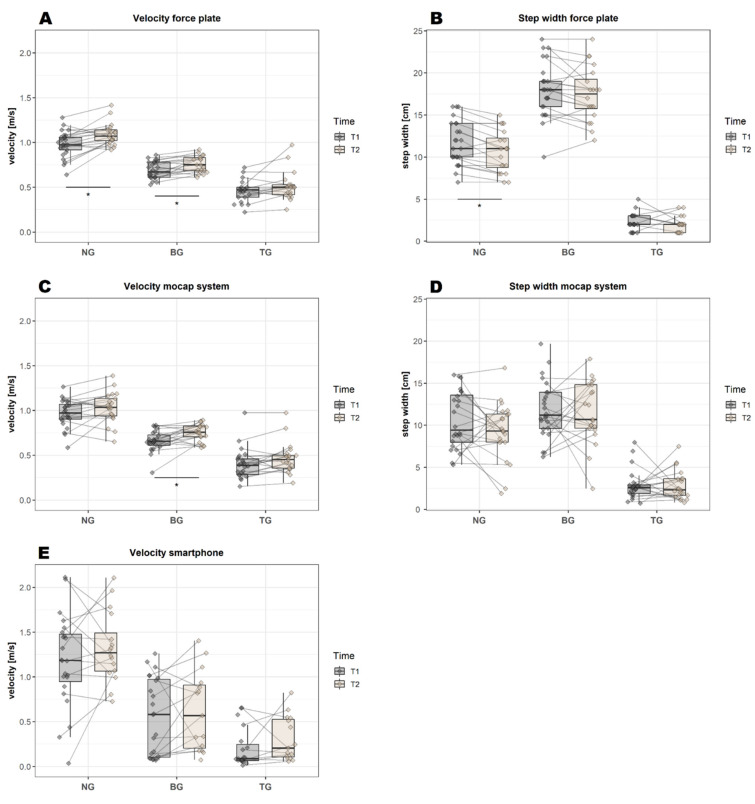
Graphical representation of the mean values of velocity and step width for all three gait analysis systems at T1 and T2 (before and after training). Significant differences over time (after Bonferroni correction) are highlighted by an asterisk. BG = backward gait, NG = normal gait, TG = tandem gait.

**Figure 5 sensors-22-04975-f005:**
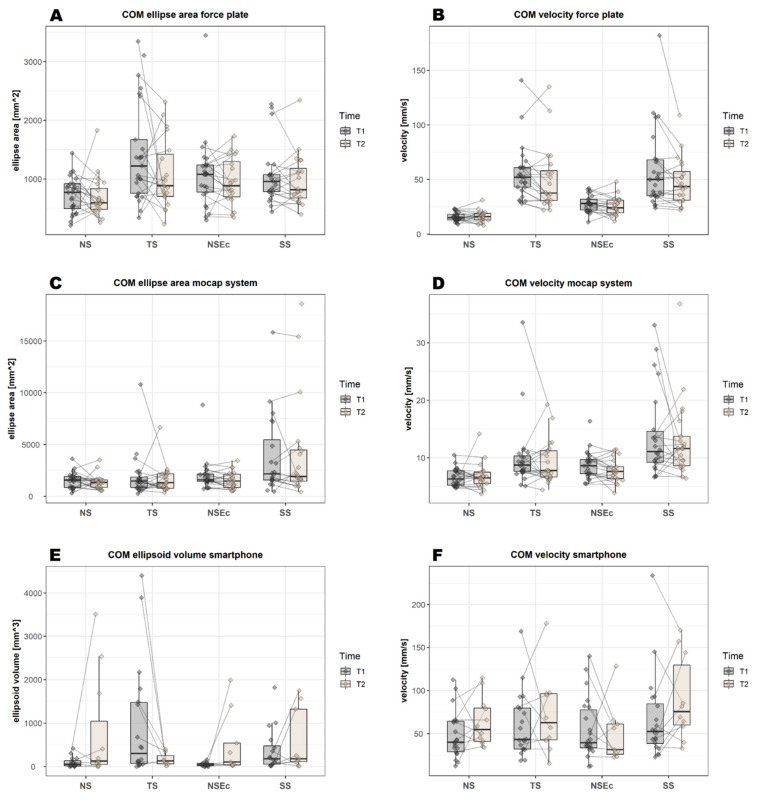
Graphical overview over the balance variables (center of mass ellipse area and velocity) in all three gait analysis systems at both measurement times (first measurement, T1, second measurement, T2). COM = center of mass, NS = narrow stance, TS = tandem stance, NSEc = narrow stance with eyes closed, SS = single leg stance.

**Table 1 sensors-22-04975-t001:** Gait and balance tasks.

Task	Content
Normal gait (NG)	10 m × 4.24 m normal (forward) gait
Backward gait (BG)	6 m × 4.24 m backward gait
Tandem gait (TG)	4 m × 4.24 m tandem gait (walk on one line placing one foot in front of the other)
Narrow stance (NS)	Balancing in a narrow stance (feet close together)
Tandem stance (TS)	Balancing in a tandem stance (feet in one line)
Narrow stance with eyes closed (NSEc)	Balancing in a narrow stance with eyes closed
Single leg stance (SS)	Balancing on one leg

**Table 2 sensors-22-04975-t002:** Overview of gait and balance variables of all gait analysis systems used for statistical analysis.

	Output Variable	Description	Unit
Gait	Stride time	Time to complete one stride (two steps)	s
	Cadence	Number of steps per second	s^−1^
	Velocity	Speed of movement	m/s
	Step width *	Lateral distance of left and right foot (center of heel) at one step	m
Balance	COM ellipse area (ellipsoid volume for smartphone)	Ellipse, enclosing 95% of all data points (100% in the mocap system) during a stance task (mediolateral and anteroposterior displacement)	mm^2^ (mm^3^)
	COM velocity	Speed of movement during a stance task (mediolateral and anteroposterior displacement)	mm/s

* not obtained with the smartphone.

**Table 3 sensors-22-04975-t003:** Demographic information of all participants (*n* = 25). Education included school years plus years up to the highest graduation achieved (e.g., German Abitur equals 12 years of education). The HADS-D anxiety score had a cut-off value of >10 and the HADS-D depression score had a cut-off value of >8. The PHQ stress score had a maximum of 20 points.

	Mean ± SD	Range (Min.–Max.)
Age [years]	44.1 ± 18.4	20–71
Body height [cm]	172.3 ± 9.9	154–193
Body weight (*n* = 17) [kg]	67.6 ± 14.2	43–97
Education [years]	15.2 ± 3.2	10–25
HADS-D Anxiety [score]	3.3 ± 2.8	0–9
HADS-D Depression [score]	2.6 ± 2.6	0–10
PHQ Stress [score]	2.8 ± 2.1	0–8

**Table 4 sensors-22-04975-t004:** Descriptive statistics of the questionnaire scores at the first and second study visit (T1, *n* = 25, and T2, *n* = 21). SE = self-efficacy (possible range: 5 to 20), OP = optimism (possible range: 2 to 8), PS = pessimism (possible range: 2 to 8). Activities-Specific Balance Confidence scale (ABC-D, possible range: 16 to 64), general habitual well-being (FAHW, average reference values between 35 and 50, smiley score ranging from 1 to 7).

	T1		T2	
Questionnaire [Score]	Mean ± SD	Range (Min.–Max.)	Mean ± SD	Range (Min.–Max.)
SWOP-SE	3.080 ± 0.49	2.0–3.8	3.229 ± 0.4485	2.2–4.0
SWOP-OP	3.240 ± 0.631	2.0–4.0	3.119 ± 0.7891	1.5–4.0
SWOP-PS	1.740 ± 0.614	1.0–3.0	1.667 ± 0.7130	1.0–3.0
ABC-D *	17.96 ± 2.574	16–28	17.76 ± 2.343	16–24
FAHW	59.12 ± 16.821	21–83	54.55 ± 25.310	−5–86
FAHW Smiley	2.04 ± 0.611	1–3	2.25 ± 0.786	1–4

* The ABC-D scores were not normally distributed. A Wilcoxon rank test was performed.

**Table 5 sensors-22-04975-t005:** Between-system correlations for normal gait between the force plate, mocap system and smartphone at T1 (first measurement time). Correlation after Pearson.

Normal Gait		Force Plate (*n* = 23)	Mocap System (*n* = 22)		Force Plate (*n* = 24)	
Smartphone	Stride time	0.977 **	0.962 **	Mocap system	Stride time	0.981 **
	Cadence	0.942 **	0.934 **		Cadence	0.992 **
	Velocity	0.705 **	0.648 **		Velocity	0.925 **
	Step width				Step width	0.430 *

* Correlation is significant at the 0.05 level (2-tailed). ** Correlation is significant at the 0.01 level (2-tailed). *n* = number of participants included in the analysis.

**Table 6 sensors-22-04975-t006:** Between-system correlations for backward gait between the force plate, mocap system and smartphone at T1 (first measurement time). Correlation after Pearson.

Backward Gait		Force Plate (*n* = 23)	Mocap System (*n* = 22)		Force Plate (*n* = 24)	
Smartphone	Stride time	0.936 **	0.706 **	Mocap system	Stride time	0.731 **
	Cadence	0.919 **	0.685 **		Cadence	0.687 **
	Velocity	0.508 *	−0.019		Velocity	0.453 *
	Step width				Step width	0.361

* Correlation is significant at the 0.05 level (2-tailed). ** Correlation is significant at the 0.01 level (2-tailed). *n* = number of participants included in the analysis.

**Table 7 sensors-22-04975-t007:** Between-system correlations for *tandem* gait between the force plate, mocap system and smartphone at T1 (first measurement time). Correlation after Pearson.

Tandem Gait		Force Plate (*n* = 17)	Mocap System (*n* = 19)		Force Plate (*n* = 19)	
Smartphone	Stride time	0.875 **	0.899 **	Mocap system	Stride time	0.901 **
	Cadence	0.794 **	0.869 **		Cadence	0.861 **
	Velocity	0.149	0.365		Velocity	0.618 **
	Step width				Step width	−0.150

** Correlation is significant at the 0.01 level (2-tailed). *n* = number of participants included in the analysis.

**Table 8 sensors-22-04975-t008:** Between-system correlations for the stance tasks at T1. Cor. = correlation after Pearson. NS = narrow stance. TS = tandem stance. NSEc = narrow stance with eyes closed. SS = single leg stance. The number of participants included in each analysis varied between 14 and 24.

			Force Plate	Mocap System			Force Plate	
Smartphone	Narrow stance	Ellipse	−0.072	0.093	Mocap system	Narrow stance	Ellipse	0.697 **
		Velocity	0.186	0.190			Velocity	0.673 **
	Tandem stance	Ellipse	0.550 *	0.315		Tandem stance	Ellipse	0.483 *
		Velocity	0.008	0.123			Velocity	0.468 *
	Narrow stance eyes closed	Ellipse	0.120	−0.058		Narrow stance eyes closed	Ellipse	0.782 **
		Velocity	0.580 *	0.210			Velocity	0.752 **
	Single leg stance	Ellipse	0.453	0.479		Single leg stance	Ellipse	0.672 **
		Velocity	0.243	0.528 *			Velocity	0.706 **

* Correlation is significant at the 0.05 level (2-tailed). ** Correlation is significant at the 0.01 level (2-tailed).

**Table 9 sensors-22-04975-t009:** Overview of values of gait variables found in the literature versus results of this study. A value description is given, unless values are mean ± SD.

		Literature			Own Results
		Values	System	Reference	(Force Plate, Mocap System, Smartphone)
Normal gait	stride time [s]	1.16 (0.92–1.41) (median (5th–95th percentiles))	zebris force plate	Pawik et al., 2021 [41]	1.18, 1.20 and 1.20
		1.09 ± 0.08	zebris force plate	Kasović et al., 2020 [42]	
	cadence [steps/s]	1.83 ± 0.17	zebris force plate	Kasović et al., 2020 [42]	1.66, 1.70 and 1.67
		1.72 ± 0.17	GAITRite force plate	Rao et al., 2011 [43]	
	velocity [m/s]	1.25 ± 0.14	zebris force plate	Kasović et al., 2020 [42]	0.98, 0.97 and 1.18
		0.94 ± 0.25	GAITRite force plate	Rao et al., 2011 [43]	
	step width [cm]	11.65 ± 2.85	zebris force plate	Kasović et al., 2020 [42]	11.64 and 10.6
		5–13 (usual walking base)		Whittle, 2007 [44]	
		11 ± 4	GAITRite force plate	Rao et al., 2011 [43]	
Backward gait	stride time [s]	1.2 ± 0.1	zebris force plate	Gimunová et al., 2021 [45]	1.22, 1.21 and 1.23
	cadence [steps/s]	1.68 ± 0.15	zebris force plate	Gimunová et al., 2021 [45]	1.66, 1.66 and 1.67
	velocity [m/s]	0.87 ± 0.12	zebris force plate	Gimunová et al., 2021 [45]	0.69, 0.66 and 0.55
		0.98 ± 0.23	GAITRite force plate	Edwards et al., 2020 [46]	
	step width [cm]	16.8 ± 4.87	zebris force plate	Gimunová et al., 2021 [45]	18.08 and 11.86
Tandem gait	cadence [steps/s]	0.8 ± 0.05 (estimated mean ± SD at 1 km/h speed)	zebris ultrasound system	Kronenbuerger et al., 2009 [47]	1.23, 1.19 and 1.23
		0.87 ± 0.29	GAITRite force plate	Rao et al., 2011 [43]	
	velocity [m/s]	0.27 ± 0.13	GAITRite force plate	Rao et al., 2011 [43]	0.45, 0.4 and 0.20
	step width [cm]	3.5 ± 2.6	GAITRite force plate	Rao et al., 2011 [43]	2.24 and 2.44

## Data Availability

The sensor data presented in this study are openly available in the pedestrian dynamics data archive “Clara Rentz, Mehran Sahandi Far, Maik Boltes, Alfons Schnitzler, Katrin Amunts, Juergen Dukart, Martina Minnerop. System comparison for gait and balance monitoring used for the evaluation of a home-based training. Forschungszentrum Jülich, 2022” at https://doi.org/10.34735/ped.2021.1, accessed on 30 June 2022.

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
