# Peer review of "System Comparison for Gait and Balance Monitoring Used for the Evaluation of a Home-Based Training"

_sensors, 2022, doi:10.3390/s22134975_

Round 1

Reviewer 1 Report

In this paper, the authors present a study about compared a smartphone-based approach with two standard gait analysis systems (force plate, and motion capturing system). The results indicate significant differences over time were found for several force-plate and mocap system obtained gait variables of normal, backward, and tandem gait. Changes in balance variables over time were more heterogeneous and not significant for any system. The smartphone seems to be an equipollent method to measure cadence and stride time of different gaits but not balance tasks in healthy adults. Additional optimizations in data evaluation and processing may further improve the agreement between the analysis systems. This article is clear, concise, and suitable for the scope of the journal. Only several small suggestions are supplied:
1. Suggest the authors supply the smartphone and the software screen if possible.
2. Suggest the authors enhance the introduction part with other smartphone-based physiological monitoring technology, such as :
Low-cost plastic optical fiber integrated with smartphone for human physiological monitoring, Optical Fiber Technology 71(1):102947,2022, etc

Author Response

We would like to thank the reviewer for his/her invaluable suggestions and for the opportunity to revise our manuscript. Answers to each of the points raised by the reviewers are given below. 

Reviewer 1 (Minor): 

In this paper, the authors present a study about compared a smartphone-based approach with two  standard gait analysis systems (force plate, and motion capturing system). The results indicate  significant differences over time were found for several force-plate and mocap system obtained gait  variables of normal, backward, and tandem gait. Changes in balance variables over time were more  heterogeneous and not significant for any system. The smartphone seems to be an equipollent  method to measure cadence and stride time of different gaits but not balance tasks in healthy adults.  Additional optimizations in data evaluation and processing may further improve the agreement  between the analysis systems. This article is clear, concise, and suitable for the scope of the journal.

Only several small suggestions are supplied: 

  1. Suggest the authors supply the smartphone and the software screen if possible.

We are not entirely sure if we understand this comment correctly. Are we supposed to provide the specific smartphone type used by each participant? All smartphones were Android smartphones, but the specific type varied. Regarding software screen - questions provided via smartphone were in German and are therefore not included in the manuscript. However, screenshots of an English version of the app are available in an article of our co-authors (Far et al., 2021, https://arxiv.org/pdf/2101.10091), already cited in our manuscript. This information regarding screenshots of the app has now been added to the manuscript as well (Material and Methods, first paragraph, line 110).

  1. Suggest the authors enhance the introduction part with other smartphone-based physiological monitoring technology, such as :

Low-cost plastic optical fiber integrated with smartphone for human physiological monitoring, Optical  Fiber Technology 71(1):102947,2022, etc 

Many thanks for pointing out this publication. We are now referencing this article in our introduction (second paragraph, line 46 to 50) and hope that the revised paragraph meets your expectations.

Reviewer 2 Report

The study investigates a very attractive possibility for those involved in clinical evaluations of both neurological and orthopedic patients. Surely the possibility of being able to record some parameters at home offers a more complete picture of the patient's progression.

However, there are some points that I would like to be clarified in the article:

The type of study should be better defined within the materials and methods.

A flowchart should be made to clarify the study design. In my opinion it is always useful to insert the part dedicated to the limits of the study.

Furthermore, the sample chosen seems very heterogeneous, it would be advisable to clarify how the constitution of the sample and its number was arrived at.

Within the discussion and in the conclusions it should be emphasized, in the light of the agreement found only on some parameters, to which type of patient and on which pathology this monitoring could be applied.

Best Regards

Author Response

Reviewer 3:

The study investigates a very attractive possibility for those involved in clinical evaluations of both neurological and orthopedic patients. Surely the possibility of being able to record some parameters at home offers a more complete picture of the patient's progression.

We thank the reviewer for the thoughtful comments. Please find point by point responses below.

However, there are some points that I would like to be clarified in the article:

The type of study should be better defined within the materials and methods.
We agree with the reviewer that the type of the study (interventional feasibility study) should be made clearer and accordingly added it to the first sentence of the Material and Methods chapter (line 105) and to the abstract (line 22).

A flowchart should be made to clarify the study design. In my opinion it is always useful to insert the part dedicated to the limits of the study.
We now provided a flow diagram (Figure 1) to facilitate the overview over the whole trial and included it into the method section. 

Furthermore, the sample chosen seems very heterogeneous, it would be advisable to clarify how the constitution of the sample and its number was arrived at.
The method of recruitment and inclusion/exclusion criteria have now been added to the manuscript (section 2.1 Participants). Anthropometric characteristics of the participants have been integrated into the Demographic Information (Table 3) in the manuscript. Also, we would like to mention that we chose a large age range to facilitate later comparison with patient groups and to focus not only on age effects of a particular group.

Within the discussion and in the conclusions it should be emphasized, in the light of the agreement found only on some parameters, to which type of patient and on which pathology this monitoring could be applied.
A paragraph was added to the Discussion (4.5 Summary, line 692-698) that describes how the results can be implemented into future research and clinical practice.

Reviewer 3 Report

I am appreciated to review the manuscript titled “System comparison for gait and balance monitoring used for the evaluation of a home-based training”. The topic of this study is attractive. However, I think some significant concerns should be identified and a substantial revision is needed. My comments are attached below.

Introduction

1.     Line 37 to 71, the author should cite some epidemiologic studies to illustrate the relationship between gait, balance, and neurological disorders.

2.     Line 62 to 67, I think the authors aimed to demonstrate the application of force plate and body-worn sensors in assessing the effect of exercise training on gait and balance, therefore, in this part, it would be better if the authors had focused on methods rather than the effects of interventions.

3.     Line 68 to 79, there are many approaches to assess the balance, the author should illustrate the “Golden standard” in clinical practice and the reliability and validity of other approaches.

4.     Line 81 to 86, before drawing this conclusion, the author should identify the advantages and differences between the two systems. Moreover, the authors should introduce their contribution to the field of their work.

5.     Why do the authors mention the COVID-19? It seemed irrelevant to the topic of this study.

Material and methods

1.     The authors didn’t illustrate their method of the recruitment and allocation of the participants, neither the inclusion/exclusion criteria of participants.

2.     Section 2.1.1, the authors should present the details of the positions in which the sensors are located and demonstrate that they have taken some necessary procedures to reduce errors such as the anthropometric differences among participants.

3.     To make the trial repeatable, the author should provide their methods of signal processing such as the frequency of the filter, smoothing, and rectification, especially the signal collected by the force plate.

4.     The author should explain why they didn’t set a blank group in their trials. Since they provided training tasks to the participants.

Results and discussion

1.     The author should provide the anthropometric characteristics of the participants.

2.     The participants took training at home via provided video, therefore, the author should provide the completion of the participants. It would be better if the authors could provide a flow diagram of the whole trial.

3.     There were many outliers in Figure 2 and Figure 3, the authors should provide the results of the normality test.

4.     In this manuscript, the authors aimed to compare a smartphone-based approach with two standard gait analysis systems (force plate, and motion-capturing system). Therefore, I cannot understand why the authors collect the patients-reported outcome measures of the participants. These questionnaires seemed irrelevant to the functions of the smart-based monitoring system.

5.     The structure of the discussion should be rebuilt since the authors didn’t focus on the reliability, validity, advantages, and limitations of the smart-based monitoring system. Therefore, the whole manuscript seems particularly long and unfocused.

6.     Considering the individual difference in the gait and balance in every single trial, I doubt that the result of the statistical analysis would be misleading since the authors used the mean value of stride time, cadence, velocity, and step width to demonstrate their conclusion. I suggested the authors compare more gait parameters in different gait phases within each participant and between groups and apply more sophisticated data analysis approaches.

Author Response

I am appreciated to review the manuscript titled “System comparison for gait and balance  monitoring used for the evaluation of a home-based training”. The topic of this study is attractive.  However, I think some significant concerns should be identified and a substantial revision is  needed. My comments are attached below.
Thank you very much for the detailed and elaborate review which helped to improve our manuscript. We hope that we sufficiently answered all the questions and issues raised in your review. A point-by-point reply is given below.

Introduction 

  1. Line 37 to 71, the author should cite some epidemiologic studies to illustrate the relationship  between gait, balance, and neurological disorders.
    Review studies have now been added to describe typical disease symptoms related to gait (Moon, 2016; Cruiz-Jimenez, 2017; Nonnekes, 2018), and further studies were chosen to highlight the overall context of age, movement disorders and gait/balance (Balestrino, 2020; Jayadev, 2013, Marsden 2018). Please find the changes in the first paragraph of the introduction, line 39-44.
  2. Line 62 to 67, I think the authors aimed to demonstrate the application of force plate and body-worn  sensors in assessing the effect of exercise training on gait and balance, therefore, in this part, it  would be better if the authors had focused on methods rather than the effects of interventions.
    Thank you for this advice. We agree that methodological aspects are important in this context. Accordingly, we expanded the description of the applied methods for the cited studies (line 61-71).
  3. Line 68 to 79, there are many approaches to assess the balance, the author should illustrate the  “Golden standard” in clinical practice and the reliability and validity of other approaches.
    The “Timed Up-and-Go Test” has now been delineated as the golden standard for clinical practice and the force plate for research. Also, an additional review article, summarizing validity and reliability of technologies for assessing balance/posture (Baker et al., 2020), has been added. See introduction, fourth paragraph, line 74-79.
  4. Line 81 to 86, before drawing this conclusion, the author should identify the advantages and  differences between the two systems. Moreover, the authors should introduce their contribution to  the field of their work.
    The advantages and differences, as well as the contribution to the field of work, were emphasized in more detail (Introduction, last paragraph, line 94-103).
  5. Why do the authors mention the COVID-19? It seemed irrelevant to the topic of this study.
    During the COVID-19 pandemic, we observed that patients frequently discontinued important appointments on site (e.g. physiotherapy). Nevertheless, we agree with you that this issue as well as the methodological advantages of using smartphone technology in general is not exclusively related to COVID-19. The corresponding sentence has been adjusted and the link to COVID-19 deleted (Introduction, second paragraph, line 46-50). 

Material and methods 

  1. The authors didn’t illustrate their method of the recruitment and allocation of the participants, neither  the inclusion/exclusion criteria of participants.
    The method of recruitment and inclusion/exclusion criteria have now been added to the manuscript. An allocation procedure was not required, since all participants underwent the same protocol (Material and Methods, section 2.1. Participants).
  2. Section 2.1.1, the authors should present the details of the positions in which the sensors are  located and demonstrate that they have taken some necessary procedures to reduce errors such as  the anthropometric differences among participants.
    We are not entirely sure which section the reviewer refers to, since section 2.1.1. describes the force plate feature extraction, which does not use sensors. In case the section regarding smartphone feature extraction was meant (section 2.1.3), an introductory sentence has been added to describe the positioning of the smartphone. Body height was required for the positioning of the sensors belonging to the Xsens mocap system and processing steps in Xsens were performed with the use of a biomechanical model. Since we investigated only one sample and used a longitudinal study design investigating the same individuals of this group again after only a short period of three weeks, changes of weight were not expected and anthropometric differences between the subjects were not considered as relevant for our study.
  3. To make the trial repeatable, the author should provide their methods of signal processing such as  the frequency of the filter, smoothing, and rectification, especially the signal collected by the force  plate.
    A short introduction has now been added to the feature extraction of the force plate (2.1.1. force plate feature extraction, line 153-158).
  4. The author should explain why they didn’t set a blank group in their trials. Since they provided  training tasks to the participants.
    Due to the pandemic and strict guidelines and requirements at the university hospital, recruitment was unfortunately limited. Nevertheless, we agree with the reviewer, that a control group is required and the corresponding measurements are currently in progress (but not yet completed). 

Results and discussion 

  1. The author should provide the anthropometric characteristics of the participants.
    Anthropometric characteristics of the participants have now been added to the demographic information (Table 3) in the manuscript. While body height was available (as required for the Xsens mocap system) for all participants, body weight was unfortunately only available in 17/25 participants. 
  2. The participants took training at home via provided video, therefore, the author should provide the  completion of the participants. It would be better if the authors could provide a flow diagram of the  whole trial.
    Thank you for the advice. We now provided a flow diagram (Figure 1) to facilitate the overview over the whole trial. In addition to that, we would like to mention that each participant confirmed to have completed all training videos. Due to ethical and data protection issues it was not possible for us to control what every participant actually did in front of the screen. Therefore, there is no further information about the actual completion (or compliance) of the single subjects of/to the study protocol. We admit that this might be regarded as a limitation of our study. Therefore, we included this information in our method section (Material and Methods, first paragraph, line 117-119), in our result section (Results, second paragraph, line 362) and addressed it as a limitation in our discussion (section 4.2 Questionnaires, third paragraph, line 575-577). 
  3. There were many outliers in Figure 2 and Figure 3, the authors should provide the results of the  normality test.
    Normality tests were performed as stated in the method section 2.3. but we agree with the reviewer that the corresponding results were difficult to find in the manuscript as they were only mentioned in the table legends if appropriate. Thus, we have now included an additional paragraph in the results section providing the results of the normality tests in detail (section 3.1 Participants, last paragraph, line 367-370).
  4. In this manuscript, the authors aimed to compare a smartphone-based approach with two standard  gait analysis systems (force plate, and motion-capturing system). Therefore, I cannot understand  why the authors collect the patients-reported outcome measures of the participants. These  questionnaires seemed irrelevant to the functions of the smart-based monitoring system.
    We aimed to investigate the feasibility of studies using smartphones on the one hand, and wanted to compare the capability of each system to depict a training effect on the other hand. A training effect gains relevance if individuals notice a benefit subjectively as well. This is also required by regulatory authorities to prove the usefulness of a specific intervention. For that reason, we used questionnaires to evaluate the subjective benefit of each participant. It is well known that physical activity positively influences well-being and we expected to see this result in our collective as well. However, our participants showed above-average well-being even before the study and this high level of well-being did not change during study participation. We regard this result as relevant and interesting, in particular its discrepancy to the observed improvement using the technical devices. If the reviewer still deems this information irrelevant, we are open to exclude the evaluation of questionnaires from the manuscript. 
  5. The structure of the discussion should be rebuilt since the authors didn’t focus on the reliability,  validity, advantages, and limitations of the smart-based monitoring system. Therefore, the whole  manuscript seems particularly long and unfocused.
    A section was added to the section “4.5 Summary” (line 680-691) to sum up advantages, limitations and literature on reliability and validity of the smartphone system (Manor et al., 2018; Silsupadol, 2017). While high reliability and validity is usually found in lab-based settings, our study rather focused on feasibility and on measuring a training effect with the smartphone. We revised and shortened our manuscript and hope that it is now more evident that the focus of our study is not primarily on reliability or validity, but on feasibility.
  6. Considering the individual difference in the gait and balance in every single trial, I doubt that the  result of the statistical analysis would be misleading since the authors used the mean value of stride  time, cadence, velocity, and step width to demonstrate their conclusion. I suggested the authors  compare more gait parameters in different gait phases within each participant and between groups  and apply more sophisticated data analysis approaches.
    It would also have been in our interest to statistically compare a larger number of gait and balance variables. However, we wanted to focus on those variables that were available for all three gait analysis systems and that are commonly used in literature. Gait phases, for example, were not reliably measurable with the smartphone and thus excluded, even though the percentage of double stance phase (for example) displays an interesting gait variable. If the reviewer has any specific suggestions regarding a different approach of data analysis, further feedback is very welcome. 

Round 2

Reviewer 3 Report

The authors have made a good revision, it shows smartphone to be an equipollent method to measure cadence and stride time of different gait. I recommend to accept.